# Recent Studies on Metal-Embedded Silica Nanoparticles for Biological Applications

**DOI:** 10.3390/nano14030268

**Published:** 2024-01-26

**Authors:** Hye-Seong Cho, Mi Suk Noh, Yoon-Hee Kim, Jayoung Namgung, Kwanghee Yoo, Min-Sup Shin, Cho-Hee Yang, Young Jun Kim, Seung-Ju Yu, Hyejin Chang, Won Yeop Rho, Bong-Hyun Jun

**Affiliations:** 1Department of Bioscience and Biotechnology, Konkuk University, Seoul 05029, Republic of Korea; joh0302@konkuk.ac.kr (H.-S.C.); yoonhees@konkuk.ac.kr (Y.-H.K.); nagu9@hanmail.net (J.N.); heu1997@konkuk.ac.kr (K.Y.); dnjzj159159@konkuk.ac.kr (M.-S.S.); vltizk0052@naver.com (C.-H.Y.); youngjkim82@gmail.com (Y.J.K.); 2Bio & Medical Research Center, Bio Business Division, Korea Testing Certification, Gunpo 15809, Gyeonggi-do, Republic of Korea; pourlady@ktc.re.kr; 3Graduate School of Integrated Energy-AI, Jeonbuk National University, 567 Baekje-daero, Deokjin-gu, Jeonju-si 54896, Jeollabuk-do, Republic of Korea; ysj__0708@naver.com; 4Division of Science Education, Kangwon National University, Chuncheon 24341, Republic of Korea; hjchang@kangwon.ac.kr

**Keywords:** surface-enhanced Raman spectroscopy (SERS), localized surface plasmon resonance (LSPR), metal nanoparticle, silica nanoparticle, core-shell structure, bio-application

## Abstract

Recently, silica nanoparticles (NPs) have attracted considerable attention as biocompatible and stable templates for embedding noble metals. Noble-metal-embedded silica NPs utilize the exceptional optical properties of novel metals while overcoming the limitations of individual novel metal NPs. In addition, the structure of metal-embedded silica NPs decorated with small metal NPs around the silica core results in strong signal enhancement in localized surface plasmon resonance and surface-enhanced Raman scattering. This review summarizes recent studies on metal-embedded silica NPs, focusing on their unique designs and applications. The characteristics of the metal-embedded silica NPs depend on the type and structure of the embedded metals. Based on this progress, metal-embedded silica NPs are currently utilized in various spectroscopic applications, serving as nanozymes, detection and imaging probes, drug carriers, photothermal inducers, and bioactivation molecule screening identifiers. Owing to their versatile roles, metal-embedded silica NPs are expected to be applied in various fields, such as biology and medicine, in the future.

## 1. Introduction

Noble metal nanoparticles (NPs) have unique properties that differ from those of their bulk counterparts and have attracted interest in various research fields and applications, including biotechnology and biomedicine [1,2]. Noble metal NPs, such as gold and silver NPs, exhibit a phenomenon known as localized surface plasmon resonance (LSPR). Tuning and controlling LSPR is possible for sensitive detection through small changes in the size, shape, composition, and interparticle spacing of the metal NPs. In addition, LSPR strongly enhances surface-enhanced Raman scattering (SERS) [3,4,5]. SERS is a highly sensitive analytical technique that detects analytes down to the single-molecule level [6,7,8,9]. However, controlling the metal NPs is disadvantageous because of their low particle stability and easy aggregation. The signal from individual metal NPs is sometimes unsatisfactory for sensitively detecting the target analyte by utilizing electrostatic and steric stabilization using ligands [10,11,12,13]. One solution to overcome these shortcomings is to embed them in silica NPs. Silica NPs can be synthesized using the Stöber method utilizing a polycondensation reaction via hydrolysis with tetraethylorthosilicate (TEOS) and ammonium hydroxide, and their size can be controlled by adjusting the concentration or temperature of the reducing agent NH_4_OH [14,15,16,17]. In addition, silica NPs can easily bind to various ligands or antibodies by modifying the surface of SiO_2_ NPs, which can be ideal templates for embedding metals owing to the following advantages. (1) Stronger optical signal: metals embedded in silica NPs enhance the SERS capability, making them useful for biosensing applications. (2) Optical tunability: the optical properties of silica NPs can be tuned by changing the type and concentration of the metal, making them useful in various optical applications. (3) Engineered fabrication: metals embedded in silica NPs can be engineered to have specific properties, making them useful in various applications. (4) Ease of handling: silica NPs are easy to synthesize, handle, and recover, making them promising materials for various applications [18,19]. Fe_3_O_4_@SiO_2_@metal has the advantage of a simple cleaning process due to its magnetic properties [20,21,22,23,24].

This review discusses recent studies that take advantage of noble-metal-embedded silica NPs. We first introduce the optical properties affected by the structure and composite of SiO_2_@metal NPs, which are modulated by the selection of a monometal and bimetal (e.g., gold, silver, and platinum). Thereafter, we summarize recent studies utilizing SiO_2_@metal NPs or their derivatives for various biological applications such as nanozymes, sensing, imaging, drug delivery, and molecule screening. The topics for developing SiO_2_@metal NPs and representative studies are summarized in Figure 1 and Table 1, respectively.

## 2. Optical Properties of Metal-Embedded Silica Nanoparticles

### 2.1. Monometal-Embedded Silica NPs

#### 2.1.1. Au-Embedded Silica NPs

Au NPs are more stable and less toxic than other metals and exhibit photothermal effects [25,26,27,28]. Au-embedded silica NPs (SiO_2_@Au) were synthesized by introducing Au NPs onto a silica template, which has many advantages [29,30]. For example, the high scattering efficiency of SiO_2_@Au NPs was investigated with Fe_3_O_4_@SiO_2_@Au NPs and compared with single Au NPs of the same size (Figure 2A). In addition, the potential of SiO_2_@Au as a near-infrared-absorbing plasmonic nanostructure was demonstrated via local field enhancement at an 800 nm LSPR wavelength [31,32]. The silica template was treated with (3-Aminopropyl)triethoxysilane (APTS) to modify its surface with amine groups. After attaching the Au seed via charge interaction, gold was grown and synthesized using a seed-mediated growth method [33]. For SiO_2_@Au@Au NPs, the size and shell thickness of the Au NPs could be controlled by adjusting the concentration of the Au precursor, temperature during synthesis, and the ratio of the ligand to the Au precursor. As the concentration of the Au precursor increases, the Au shell thickness increases, the absorbance increases from 570 to 630 nm, and the wavelength band broadens [34]. When synthesizing SiO_2_@Au@Au NPs, the coverage increases with increasing temperature [35]. By adjusting the ratio of citrate to the Au precursor, the smaller the concentration of citrate, the larger the size of the Au shell, and the lower the coverage. Absorbance may change depending on the shell size and coverage [36]. These properties allow its utilization in many applications, including drug release, photothermal cancer treatment, diagnosis, and in vivo imaging [33,37,38,39,40].

#### 2.1.2. Ag-Embedded Silica NPs

Ag NPs have the advantage of exhibiting strong LSPR and high SERS activity compared to other metals. Ag NPs can also be introduced on silica templates to synthesize Ag-embedded silica NPs (SiO_2_@Ag NPs). Although the external coating of Ag is sometimes difficult to control, particularly in large-scale synthesis, the scaling up of stable and highly reproducible SiO_2_@Ag NPs has been proven by SERS-based studies (Figure 2C). In addition, the plasmonic light enhancement of the SiO_2_@Ag NPs between the gaps was investigated through simulation to analyze both the size and gaps between the NPs (Figure 2B) [41,42]. The synthesis was performed by treating the silica template with (3-Mercaptopropyl)trimethoxysilane (MPTS) to modify the silica surface with thiols, followed by the addition of an Ag precursor and a reducing agent. Reduced Ag is embedded in the silica NP surface due to its strong affinity with thiols [43,44]. For SiO_2_@Ag NPs, the nanoparticle size and shell thickness can be controlled by adjusting the concentration of the Ag precursor and the temperature during synthesis. In both methods of surface modification of SiO_2_ NPs with thiols or aldehydes to form Ag shells, the shell thickness can be controlled by adjusting the concentration of Ag precursors [45]. The higher the Ag precursor concentration, the thicker the shell. The thicker the shell, the wider the wavelength range. The synthesis process, when conducted at elevated temperatures ranging from 300 to 800 °C, results in an increase in the size of Ag NPs. This temperature-dependent variation is evidenced by a broader wavelength profile, showing expansion in the size of Ag NPs [46]. With these characteristics, SiO_2_@Ag NPs can be utilized in various fields, such as in detecting drugs and metabolites [47], tracking animal cells with near-IR SERS nanoprobes [43], and detecting harmful substances [48,49].

### 2.2. Bimetal-Embedded Silica NPs

Bimetallic NPs, which comprise two different metals, exhibit blended properties controlled by the type and proportion of metals used in the synthesis [50]. A well-known method for synthesizing bimetallic NPs involves the following steps: (1) co-reduction to produce alloy NPs by simultaneously reducing two metals with a reducing agent; (2) seed-mediated growth to produce core-shell particles; (3) removal of core particles of a core-shell structure to create hollow structures through anode melting; and (4) laser ablation which utilizes a laser to convert bulk bimetallic particles into NPs [50]. Among these four methods, co-reduction and seed-mediated growth are widely utilized to introduce bimetallic shells onto silica cores. Sapkota et al. developed a method for synthesizing SiO_2_@Au–Ag nanocomposites via direct reduction [51]. Pham et al. developed SiO_2_@Au-Ag NPs with Au-Ag bimetallic shells utilizing a seed-mediated growth method, and the thickness of the shell was controlled by adjusting the amount of the precursor AgNO_3_ [52]. The simulation results indicate that the SERS enhancement of Au@Ag NPs depends on the Ag shell thickness and is stronger than that of the Au NPs (Figure 3B). This advantage of metal alloys also contributes to SERS enhancement in SiO_2_@Au–Ag NPs. For SiO_2_@Au@Pt NPs, the ratio of the two metals can be adjusted by increasing the ratio; for example, the ratio of Pt can be increased by increasing the concentration of Pt ions. Due to these characteristics, active research is being conducted in the field of catalysts, such as oxygen reduction reactions (ORRs) and nanozyme [53,54,55]. Ag nanoshells (AgNSs) were prepared via the seedless and rapid growth of Ag shells on silica NPs. Au/Ag hollow nanoshells (AuHNSs) were synthesized from AgNSs through a galvanic replacement reaction. Figure 3 presents the synthesis scheme and TEM images of the Au/Ag-alloyed hollow shells [56]. AuHNSs were modified with poly(ethylene glycol) derivatives to allow the conjugation of the epidermal growth receptor (EGFR) antibody and increase biocompatibility. Next, the EGFR antibody was conjugated through activation of the carboxyl groups on the surface of the PEGylated AuHNSs, followed by doxorubicin (DOX) loading [57].

**Table 1 nanomaterials-14-00268-t001:** Representative studies utilizing metal-embedded silica NPs.

Metal-Embedded Silica NPs		Composition		Preparation					Optical Properties		Applications	Reference
		Silica Core (Size)	Metal NP (Size)	Methods	Functionalization	Growth Solution	Reducing Agent	Capping Agent	Absorbance	SERS Enhancement Factor		
Au-Embedded Silica NPs	SiO_2_@Au	Silica NPs (227 nm)	Au NPs (14 nm)	direct deposition	APTS	TurkevichAu NPs	-	-	~540 nm	-	method development	
			Au NPs	direct reduction	APTS	trisodium citrate-HAuCl4 solution	trisodium citrate	trisodium citrate	~538 nm(pH = 3.85–5.38)	-		[29]
			Au NPs	seed-mediated growth	APTS	TurkevichAu NPs	trisodium citrate	trisodium citrate	~551 nm	-		
	SiO_2_@Au	Silica NPs (600 nm)	Au NPs (10 nm)	direct deposition	APTS	Au NPs *	-	-	-	-	nitrogen adsorption	
			Au NPs (50 nm)	direct reduction	APTS	K_2_CO_3_–HAuCl_4_ solution	formaldehyde	PVP	-	-		[30]
	SiO_2_@Au	Silica NPs (100 nm)	Au NPs (1.5–40 nm)	direct reduction	amine-grafted Silica NPs *	K_2_CO_3_–HAuCl_4_ solution	NaBH_4_	sodiumcitrate dihydrate	518–634 nm(pH = 3.09–10.60)	-	photothermal conversion	[27]
	SiO_2_@Au	Silica NPs (132 nm)	Au NPs (5 nm)	seed-mediated growth	APTS	K_2_CO_3_–HAuCl_4_ solution	NaBH_4_	sodiumcitrate dihydrate	-	-	photothermal conversion	[28]
	SiO_2_@Au@Au	Silica NPs (150 nm)	Au NPs (3 nm)	seed-mediated growth	APTS	THPC Au NPs	ascorbic acid	PVP(Mw 40,000)		3.8 × 10^6^	SERS imaging	[33]
	SiO_2_@Au	Silica NPs (160 nm)	Au NPs (1–15 nm)	direct reduction	APTS	HAuCl_4_ solution	ascorbic acid	PVP(Mw 40,000)	543–632 nm	-	nanozyme	[34]
	SiO_2_@Au	Silica NPs (670 nm)	Au NPs (16–20 nm)	direct deposition	APTS	TurkevichAu NPs	-	-	~550 nm	-	method development	
				direct reduction	APTS	HAuCl_4_ solution	trisodium citrate	trisodium citrate	-	-		[35]
	SiO_2_@Au	Silica NPs (120 nm)	Au NPs (21–39 nm)	direct reduction	APTS	trisodium citrate-HAuCl4 solution	NaBH_4_	trisodium citrate	631–784 nm	2.0 × 10^5^	SERS probe development	[36]
	SiO_2_@Au@GO	Silica NPs (220 nm)	Au NPs (1–5 nm)	direct deposition	APTS	THPC Au NPs	-	-	~562 nm	-	photothermal therapy	[38]
	SiO_2_@Au	Silica NPs (400 nm)	Au NPs (15 nm)	direct deposition	APTS	TurkevichAu NPs	-	-	~523 nm	-	photothermal therapy	[40]
Ag-Embedded Silica NPs	SiO_2_@Ag	Silica NPs (150 nm)	Ag shell thickness (32–76 nm)	direct reduction	MPTS	AgNO_3_ solution	octylamine	PVP(Mw 40,000)	560–1000 nm	6.4 × 10^5^	NIR-SERS probe	[43]
	SiO_2_@Ag_RLC_-Ag	Silica NPs (150 nm)	Ag NPs (9–15 nm)	direct reduction	MPTS	AgNO_3_ solution	octylamine	PVP(Mw 40,000)	400–800 nm	1.7 × 10^7^	detection of cancer biomarker	[44]
	SiO_2_@Ag	Silica NPs (300 nm)	Ag NPs	seed-mediated growth	APTS and GA	AgNO_3_ solution	triethanolamine	-	410 nm	-	electrically conductive adhesives	[45]
	SiO_2_@Ag	Silica NPs (670 nm)	Ag NPs (10–61 nm)	powderization and heat treatment	-	AgNO_3_– NH_4_OHsolution	-	-	403–410 nm	-	method development	[46]
	SiO_2_@Ag	Silica NPs (182 nm)	Ag shell thickness (215–363 nm)	seed-mediated growth	-	[Ag(NH_3_)_2_]^+^solution	-	PVP(Mw 40,000)	436–443 nm	-	detection of drug and metabolite	[47]
	SiO_2_@Ag	Silica NPs (155 nm)	Ag NPs (9 nm)	direct reduction	-	AgNO_3_ solution	NaBH_4_	PVP	411 nm	-	heavy metal detection and catalytic activity	[48]
	SiO_2_@Ag	Silica NPs (300 nm)	Ag NPs (40 nm)	direct reduction	-	[Ag(NH_3_)_2_]^+^solution	PVP	PVP	-	1.63 × 10^6^	detection of antibiotic residue	[49]
Bimetal-Embedded Silica NPs	SiO_2_@Au@Ag	Silica NPs (150 nm)	Au@Ag NPs (11–63 nm)	seed-mediated growth	APTS	AgNO_3_ solution	ascorbic acid	PVP(Mw 40,000)	400–800 nm	4.2 × 10^6^	SERS probe development	[52]
	SiO_2_@Au@Pt	Silica NPs (160 nm)	Au@Pt NPs	seed-mediated growth	APTS	AgNO_3_ solution	ascorbic acid	PVP(Mw 40,000)	300–800 nm	-	nanozyme	[53]

* Commercial NP products were utilized in these studies. Abbreviations: aminopropyltrimethoxysilane or aminopropyltriethoxysilane (APTS); glyoxalic acid (GA); graphene oxide (GO); (3-Mercaptopropyl)trimethoxysilane (MPTS); polyvinylpyrrolidone (PVP); surface-enhanced Raman spectroscopy (SERS); and tetrakis(hydroxymethyl)phosphonium chloride (THPC).

## 3. Biological Applications of Metal-Embedded Silica Nanoparticles

### 3.1. Nanozyme

Nanozymes have several advantages over natural enzymes, including high stability in harsh environments, low production costs, and large specific surface areas [58]. Nanozymes are novel functional nanomaterials that, unlike natural enzymes made of proteins, have the same catalytic activity by synthesizing inorganic substances and can customize their catalytic activity according to size, shape, and composition. Nanozymes are used in many fields such as biofuel cells, hydrogenation, air purification, anti-aging therapy, and cancer treatment [59,60,61,62]. The types of nanozymes include both Type 1 and Type 2 nanozymes. Nanozymes utilizing Type 2 nanomaterials are being researched more actively than Type 1 nanozymes [63]. To realize the advantages of nanoparticles as nanozymes for various applications, it is important to synthesize fine NPs precisely. For the SiO_2_@Au@Pt NPs, the silica core was aminated with APTS, and Au seeds were introduced onto the aminated surface. Subsequently, they were synthesized by adding a Pt precursor. The particle size was controlled by varying the concentration of Pt^2+^. Au@Pt NPs have extensive catalytic properties, as they can effectively scavenge superoxide free radicals or formic acid [53,64]. Pham et al. developed SiO_2_@Au@Ag and SiO_2_@Au@Au NPs to detect hydrogen peroxide (H_2_O_2_) (Figure 4A) [65]. The detection of H_2_O_2_ was evaluated between 40 and 100 mM, and the LOD was 33.3 mM. Later, Pham et al. also developed SiO_2_@Au@Pt, which has more stability and better catalytic performance [53]. The detection of H_2_O_2_ was conducted between 1.0 and 100 mM, and the LOD was 1.0 mM. Figure 4B demonstrates that TMB was oxidized to oxTMB, which exhibits peroxidase activity [34]. In spite of its possibility, there is not much research yet on metal-embedded silica NP-based nanozymes. We think that the enhancement of the efficiency and functionality of nanozymes by introducing various metals into silica NPs would be a fascinating issue to address in nanozyme research. Table 2 below is a table summarizing information about nanozymes.

### 3.2. Sensing and Detection

Metal-embedded silica NPs exhibit unique plasmonic characteristics derived from embedded metal NPs. In addition, the abundant hotspots make them promising SERS probes with strong and uniform signals. Therefore, they have been utilized in various studies to detect tiny amounts of targets of interest, such as disease-related molecules and toxic substances. This subsection summarizes the representative applications of metal-embedded silica NPs in sensing.

#### 3.2.1. Hazardous Substance Detection

Histamine is an organic, nitrogenous substance involved in immune responses, physiological functions, and neurotransmission. As seafood products with high histamine levels cause allergic responses, the rapid detection of histamine is an important issue in food safety and public health. Huynh et al. developed metal-embedded silica NPs (SiO_2_@Au@Ag NPs) as SERS probes for histamine detection [66]. The authors first optimized the quantity of NPs and found that the SERS signal decreased as the quantity of particles increased. The bimetallic metal enhanced the signal, and the limit of detection (LOD) was 0.033 mM (3.698 ppm) under optimized conditions. The LOD is considerably lower than the U.S. Food and Drug Administration and the European Union regulations.

Likewise, antibiotics or their residue cause serious concerns in the environment. For example, amoxicillin is a penicillin antibiotic that fights against bacterial infection. However, its effluent diffused in the water destroys ecological balance. Guo et al. prepared a silver-embedded silica NP (SiO_2_@Ag) as a SERS substrate to detect amoxicillin sensitively and specifically [49]. The LOD was 2.7 × 10^−7^ M with the composites themselves and 2.7 × 10^−9^ M with the modification of receptors on the composites.

Another example is the toxic compound 4-aminophenol (4-AP). It is a major impurity of N-acetyl-p-aminophenol (APAP), one of the most common drug ingredients. 4-AP is produced during the synthesis and storage of APAP. However, it has toxicity and is a cause of nephrotoxicity and hepatotoxicity. Pham et al. detected 4-AP in APAP utilizing metal-embedded silica NPs (SiO_2_@Au@Ag) [67]. 4-AP can bind to the surface of the metal, whereas APAP has no affinity. Consequently, a characteristic peak of 4-AP at 1591 cm^−1^ was confirmed after incubation (Figure 5B). Interestingly, the presence of APAP significantly affected the SERS signal. Therefore, the authors could successfully detect the production of impurities. The LOD was 3.5 ppm, much lower than the maximum allowable concentration suggested by the British Pharmacopoeia (approximately 1000 ppm).

Thiram, a component of pesticides, is utilized to prevent fungal diseases in fruits, vegetables, and ornamentals; however, it persists after harvesting and can be harmful to human health. In recent study, thiram was successfully detected with a label-free method utilizing a SiO_2_@Ag nanoshell (AgNS) as the SERS-active nanostructure by Yang et al. [68]. The synthesis of AgNSs was optimized with 3.5 mM AgNO_3_, 5 mM octylamine, and 0.06 mg/mL of silica NPs. The detection of thiram with the optimized AgNS was measured from thiram concentrations of 10^−1^ M–10^−8^ M. Following the adsorption of various concentrations of thiram into the apple peel, SERS indicated a tendency to increase as the concentration of thiram increased (Figure 5A). Detection of thiram can be confirmed at a level of 38 ng/cm^2^, which is much lower than the maximum allowable concentration of 2 μg/cm^2^ in apple skin.

Heavy metals are also one of the major contaminants that are emerging as a serious environmental problem. Kim et al [69]. performed the detection of aqueous mercuric ion (Hg^2+^) by utilizing the amalgamation characteristics of Ag NP-embedded silica NPs (SiO_2_@Ag) as SERS substrates. Following the amalgamation of SiO_2_@Ag by simply mixing with aqueous Hg^2+^, the Ag NPs on the silica NP core coalesce each other and make their electromagnetic hotspots deform. This morphological change is accompanied by a significant decrease in the SERS signal of 4-fluorothiophenol adsorbed on SiO_2_@Ag, enabling the simple and reproducible detection of Hg^2+^ [69]. Khedkar et al. also reported the synthesis of silver-embedded silica NP (SiO_2_@Ag) to detect mercury ions [48]. The nanocomposites successfully prepared via a simple chemical route interact with Hg^2+^ with a detection limit of 0.9 μM (0.2 ppm). Interestingly, the authors further investigated the catalytic activity of the nanocomposites for treating dye molecules.

#### 3.2.2. Biomarker Detection

Glucose detection is one of the most important areas of biosensor research. Monitoring glucose levels is important for managing and treating diabetes [70,71,72]. Pham et al. utilized metal-embedded silica NPs (SiO_2_@Au@Ag NPs) containing 4-mercaptophenyl boronic acid (4-MPBA) as the Raman labeling compound (RLC) in conventional glucose sensing based on glucose oxidase (GOx). The H_2_O_2_ produced by the action of GOx was measured by converting 4-MPBA to 4-mercaptophenol (4-MPhOH) [71]. Because 4-MPBA was converted into 4-MPhOH in the presence of H_2_O_2_, changes in the signal were measured using SERS. Under the optimized condition, the limit of detection (LOD) was estimated to be 0.15 mM, and the linear range ranged from 1.0 to 8.0 mM.

The advantages of utilizing metal-embedded silica NPs as sensing probes are also useful for immunoassay-based applications. Immunoassays are simple, efficient, and inexpensive. Hence, they are a representative form of lab-based testing [73,74,75]. The most widely utilized immunoassays are radiolabeled immunoassay (RIA), enzyme-linked immunosorbent assay (ELISA), lateral flow immunoassay (LFIA), and SERS-based immunoassay (SIA) [76,77,78]. When metal-embedded silica NPs are introduced into these immunoassay systems, we can utilize a convenient immunoassay format with enhanced sensing performance. Chang et al. reported the SIA-based detection of prostate-specific antigen (PSA) using metal-embedded silica NPs (SiO_2_@Ag@SiO_2_) [79]. PSA is a widely utilized biomarker for the early diagnosis of prostate cancer, a common cancer in men worldwide. Prostate cancer ranks fifth in cancer-related mortality [80,81]. As demonstrated in Figure 6A, 4-fluorobenzenethiol (4-FBT) was attached to the Ag surface, and the NPs were covered with a silica shell (~30 nm). SIA was conducted using this antibody-conjugated SERS dot, and the entire chip was analyzed with a full-area raster scanning method with a micro-Raman system. Based on the sensitive and photostable SERS dots, the proposed system demonstrated the ability to detect single-particle levels. The LOD was calculated to be 0.11 pg/mL, and the dynamic range was five orders of magnitude.

Cancer biomarkers are the most important field in immunoassays. Kang et al. presented a silver-embedded silica NP (SiO_2_@Ag) to perform an SIA-based detection of PSA and further evaluations of multiplex-capable SERS tags in bioimaging applications [44]. The authors utilized a non-small cell lung cancer cell line (H522) to evaluate the expression of human epidermal growth factor-2 and epidermal growth factor receptor.

Pham et al. proposed another SIA-based detection method utilizing metal-embedded silica NPs (SiO_2_@Au@Ag) and liposomes [82]. Liposomes are small lipid vesicles comprising amphiphilic molecules. They have many advantages, including biocompatibility, large surface area, encapsulation ability, and easy surface functionalization [83,84]. Because of their large internal volumes, liposomes are considered ideal drug carriers in drug-carrier systems. The authors prepared adenosine triphosphate (ATP)-encapsulated liposome and metal-embedded silica NPs to detect the signal from ATP during liposome lysis. The LOD of the liposome was calculated to be 1.3 × 10^−17^ mol with a linear range between 8 × 10^6^ and 8 × 10^10^ mol.

**Figure 6 nanomaterials-14-00268-f006:**
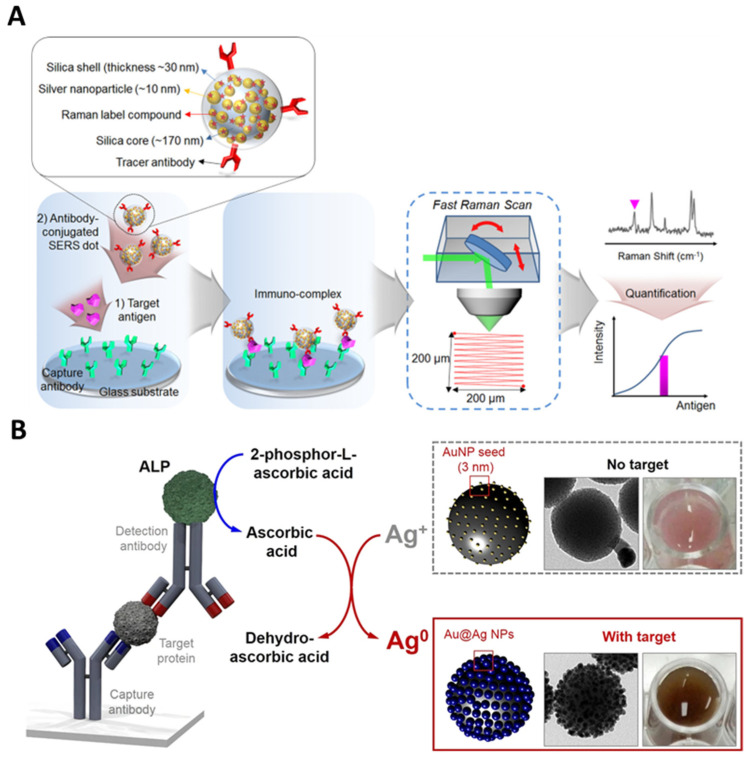
Indirect detection for immunoassay methods. (**A**) Schematic illustration of a chip-based immunoassay using SERS dots and area-scanning readout system, adapted from Chang H et al., 2016 [79]. (**B**) Schematic illustration of enzyme-catalyzed Ag growth on Au NP-assembled structures. In the presence of the target IgG, the immune complex Ab_1_-IgG-Ab_2_, which contains alkaline phosphatase (AP), triggers the enzyme-catalyzed conversion of 2-phospho-L-ascorbic acid to ascorbic acid, adapted from Pham et al., 2018 [85].

Pham et al. developed a sensitive colorimetric immunoassay utilizing metal-embedded silica NPs (SiO_2_@Au@Ag) [85]. To amplify the sensitivity of ELISA analysis, the authors conducted enzyme-catalyzed Ag growth on the Au surface. As demonstrated in Figure 6B, alkaline phosphatase (AP) was conjugated to the detection antibody to convert 2-phospho-L-ascorbic acid to ascorbic acid. This indicated that the reaction of the detection antibody in the presence of the target molecules increased the concentration of ascorbic acid in the substrate. Therefore, Ag ions were reduced, and the growth onto SiO_2_@Au proceeded. The absorbance varied depending on the Ag concentration, enabling a highly sensitive colorimetric immunoassay.

LFIA is a paper-based platform for detecting and quantifying analytes in complex mixtures, with results displayed within 30 min. Because of its simplicity, it is popular in various fields, including the biomedical, agricultural, food, and environmental sciences [86,87,88]. Although it is a rapid and inexpensive platform that enables portable detection, its sensitivity is often unsatisfactory for the sensitive and quantitative detection of the target analyte. Metal-embedded silica NPs can be a solution to overcome these issues.

Kim et al. developed a semiquantitative LFIA for PSA detection using metal-embedded silica NPs (SiO_2_@Au-Ag) [89]. The scattering effect of the proposed NPs containing bimetallic components was superior to that of single metal-embedded silica NPs (SiO_2_@Au) and Au NPs. PSA levels were measured by combining SiO_2_@Au-Ag and LFIA with a reporter. Owing to the interaction between PSA, the anti-PSA antibody-conjugated SiO_2_@Au-Ag NP complex, and the antibody, the test line turned dark brown within 15 min, and the signal intensity of the test line increased with increasing PSA concentration (Figure 7).

The role of LFIA has been highlighted during the coronavirus disease 2019 (COVID-19) pandemic [90]. Research on the rapid and accurate diagnosis of COVID-19 is being actively conducted. Hong et al. developed an LFIA-based method for detecting nucleocapsid proteins in severe acute respiratory syndrome coronavirus 2 (SARS-CoV-2) utilizing metal-embedded silica NPs (SiO_2_@Au) [74]. The authors designed NPs with core-satellite structures for strong absorption and weak scattering. Under conditions optimized to maximize specific binding and minimize nonspecific binding, the proposed system could detect 1 pg/mL of nucleocapsid protein within 20 min. The combination of LFIA and SiO_2_@metal NPs is expected to be a benchmark for improved diagnostic technologies. Currently, the sensing and detection application of silica-based metal nanoparticles is the most actively pursued. We think that more research will be conducted in the future because it has advantages in signal improvement and stability compared to single nanoparticles. Table 3 below is a table summarizing information about sensing and detection.

### 3.3. Bioimaging

Silica NPs possess several essential characteristics for imaging probes, including biocompatibility and stability. Regarding the problems associated with single metal-NP-based imaging applications, such as low reproducibility, uncontrollable aggregation, and background signals, metal-embedded silica NPs represent one of the most promising solutions. The simultaneous fluorescence Raman endoscopic system (FRES) detects and images molecular changes in tissues in real time during endoscopy, as seen in Figure 8A [91,92,93]. FRES comprises the following three components: a dual-axis laser scanning device capable of accepting incident light and collecting data on the entire fiber bundle, a fluorescence and SERS signal separation device for the simultaneous detection of both signals of target F-SERS dots, and a photodiode for fluorescence imaging and SERS spectral measurements. The optical beam guards of the fluorescence and Raman signals, having two detection units comprising a spectrometer with a CCD camera, were completely separated and detected independently. FRES can simultaneously detect fluorescence images and Raman spectra in real time, utilizing a single laser. It employs F-SERS dots, comprising particles wherein a fluorescent dye is bound to SiO_2_@Ag-RLC@SiO_2_ NPs. This method utilizes the fluorescence signal emitted by the AF610 dye to trace the location of F-SERS dots in real time. Further experiments utilizing FRES should investigate the long-term toxicity of F-SERS dots and confirm the minimum dose [94].

The SERS dots can be detected at the single-particle level utilizing NIR, with a signal strength 100 times stronger than single AuNPs (80 nm), as shown in Figure 8B. In addition, it provides cost-effective multimodal measurements for developing antibody-based drugs. To improve the sensitivity of SERS nanoprobes during in vivo applications, the RLC electron resonance by near-infrared excitation, shape of NPs, LSPR of the substrate, optical tuning, and shape modification of the LSPR have been investigated. To validate the performance of NIR SERS dots, TSPAN2 antibodies against human colorectal cancer were tested and revealed significant results [95].

Du et al [96]. fabricated Au NP-embedded silica NPs (SiO_2_@Au nanoshells) for laser desorption/ionization mass spectrometry (LDI-MS) and mass spectrometry imaging (MSI), as shown in Figure 9. LDI-MS irradiates a laser onto the surface of nanoparticles, evaporates them, and turns them into an ionized state. LDI-MSI using SiO_2_@Au nanoshells is promising for mapping the distribution of small molecules and lipids with an excellent LOD. The authors overcame the background interference issue in conventional LDI-MSI, which is derived from commonly utilized organic matrices, by adapting SiO_2_@Au nanoshells. The nanoscale roughness of SiO_2_@Au nanoshells enhanced plasmonic effects, while the crevice space acted as a trapping site for small molecules or cations. Based on these synergistic effects, the authors detected small molecules, including amino acids, carbohydrates, dyes, and drugs, with LOD ranging from 1 pmol to 1 fmol. They also demonstrated tissue imaging with strawberry fruit, zebrafish, honeybee, and mouse brain tissue with pixel sizes of 100, 55, 30, and 50 μm, respectively. For example, the analysis of strawberry metabolites clearly visualizes the tissue-specific distribution of metabolites, such as sugars, amino acids, organic acids, and anthocyanins. The mapping of zebrafish visualizes the distribution of a wide range of lipid species, including phosphatidylcholine (PC), lyso-PC (LPC), diglycerides (DG), and triglycerides (TG) throughout the body. The mapping of honeybees confirmed that most TG species, such as TG (48:2), TG (51:2), and TG (54:3), are distributed in the brain tissues of bees. Some TG species, including TG (53:4) and TG (51:3), were distributed in the systemic tissues, whereas most PC species, including PC (36:1), PC (34:2), and PC (34:1), were distributed in the muscle tissue. The lipid distribution analysis in mouse brain tissue indicated a predominant presence of PC, particularly PE (42:7), PE (37:1p), phosphatidyl glycerol (PG) (38:3), PG (38:4), and PG (38:5). The distribution was confirmed specifically in the hippocampus, demonstrating the visualization capability of lipid distribution in mouse brain tissue [96]. In bioimaging, the specific attachments of imaging probes on the target cells or lesions are the most important functions. For these reasons, research is being conducted using various ligands, such as antibodies and peptide aptamers, which take advantage of the ease of surface modification of silica NPs. Although metal-embedded silica NPs are known to be biocompatible, there are some limitations, especially in terms of in vivo toxicity. We think that much research on toxicity will be needed in the future for practical application. Table 4 below is a table summarizing information about bioimaging.

### 3.4. Drug Carriers and Photothermal Therapy

Nanocarrier-based drug delivery offers excellent local accessibility and bioavailability. One advantage of silica NPs is their large surface area and controllable porosity. Therefore, they are promising support materials for drug carriers. Metal-embedded silica NPs possess multifunctional potential, notably in applications such as hyperthermia using the photothermal effect. The photothermal effect of metal nanoparticles occurs when light is irradiated on the plasmonic metal, resulting in LSPR. It has a photothermal effect due to thermal energy generated by hot electrons that occurs thereafter. The converted heat increases local temperature, increasing treatment efficacy and reducing normal tissue damage by injecting plasmonic nanoparticles into the affected area and then irradiating the laser [57,97,98,99,100,101].

Noh et al. developed Au/Ag hollow nanoshells, designed to serve as versatile drug carriers that can target cancer cells and release drugs via the photothermal effect [57,97,98,99,100]. The authors investigated the loading and release properties of a model drug (DOX) into AuHNS NPs under various conditions. They proved that AuHNS-EGFR-DOX can kill specific cancer cells via effective drug delivery with specific targeting. The therapeutic efficacies of the PEGylated AuHNSs for lung cancer were compared with two targeting methods: maximizing the ability to target cancer cells, drug release, and thermotherapy via NIR irradiation (Figure 10).

The silica capping on the outside of metal-embedded silica NPs is an important strategy to lower the toxicity of the particles in therapeutic applications. Park et al. synthesized Au-embedded silica NPs (SiO_2_@Au@SiO_2_) to treat human mesenchymal stem cells (hMSCs) [40]. The authors found that SiO_2_@Au@SiO_2_ had no cytotoxicity below a concentration of 20 μg/mL. In the further investigation of tumor-bearing mice, SiO_2_@Au@SiO_2_ showed a higher in vivo photothermal effect compared to the individual AuNPs or gold-embedded silica NP (SiO_2_@Au).

Cyclodextrin (CD) is commonly used for drug delivery because of its hydrophobic inner surface. Kang et al. synthesized β-CD derivatives on metal-embedded silica NPs (SiO_2_@Ag) [102]. With cysteine-β-CD and ethylenediamine-β-CD (EDA-β-CD) binding to SiO_2_@Ag, the proposed system works as a nanocarrier that can release the drug in controllable ways. The authors verified the non-cytotoxic nature of β-CD and its proficiency in loading DOX. Subsequently, upon releasing the captured DOX into cancer cells, the viability of cancer cells decreased in accordance with the release time of DOX.

Graphene oxide (GO), which has a high specific surface area, also helps to increase the drug-loading capacity of NPs. Xiaolin et al. fabricated a drug carrier/photothermal inducer utilizing metal-embedded silica NPs (SiO_2_@Au@GO) [38]. GO indicated a high loading capacity for the aromatic drug via π–π stacking in the multifunctional antitumor particle with a core–shell structure. The authors confirmed the loading capacity and pH-dependent release of the model drug docetaxel (Dtxl). When human prostate cancer DU145 was cultured on SiO_2_@Au@GO-Dtxl and irradiated with 780 nm NIR, cell viability decreased owing to the photothermal effect. In addition, when cultured at pH 7.4 and 5.5, it was confirmed that the cell viability decreased to a more acidic pH of 5.5. Currently, research is being conducted under various conditions, such as the type and shape of metal, and cancer therapy using the EPR effect and photothermal advantages, which are characteristics of nanoparticles, is also being researched. In addition, research is being conducted to improve drug loading capacity by modifying the surface of silica, selecting core material, and/or using mesoporous silica. We believe that it will be one of the actively studied applications in the future due to its potential. Table 5 below is a table summarizing information about drug carriers and photothermal therapy.

### 3.5. Molecule Screening

For high-throughput bioanalysis, a method for directly identifying peptides on beads utilizing a surface-enhanced Raman spectroscopy barcode system was investigated. A simple SERS barcode method utilizing highly sensitive SERS nanoidentifiers (SERS ID) was investigated. The 44 types of SERS IDs could generate simple codes and possibly more than one million types of codes by incorporating combinations of different SERS IDs. The barcoding method exhibited high stability and reliability under the bioassay conditions. The SERS ID encoding-based screening platform identified the peptide ligand on the beads and quantified its binding affinity for specific proteins. Figure 11 shows a schematic diagram of the peptide-encoding process with SERS ID [103].

## 4. Conclusions

When noble metal NPs are introduced into a silica template, the resulting NPs exhibit combined properties of noble metal NPs and silica cores. These metal-embedded silica NPs make use of the excellent and unique properties of noble metal NPs while overcoming limitations such as low particle stability and easy aggregation of individual metal NPs. Owing to the inert and versatile properties of silica templates, metal-embedded silica NPs are advantageous for controlling their size and surface properties. The LSPR signal of the metal-embedded silica NPs was stronger than that of individual metal NPs. In addition, metal NPs assembled on the silica core generated enhanced SERS signals. However, we think that issues such as the toxicity and emission of nanoparticles in vivo and the impact of nanoparticles when they enter the environment should be continuously improved, and research into problem solving should continue.

Nevertheless, based on the above advantages, we anticipate that the scientific interest in metal-embedded silica NPs will keep growing. As we described above, metal-embedded silica NPs play various roles in spectroscopic applications, such as nanozymes, detection probes, imaging probes, drug carriers, photothermal inducers, and screening identifiers for bioactivation molecules. Although further investigation will be required to understand their influences on toxicity in vivo, the field of application of these potentially valuable NPs will keep expanding, especially in the biological and medical fields.

## Figures and Tables

**Figure 1 nanomaterials-14-00268-f001:**
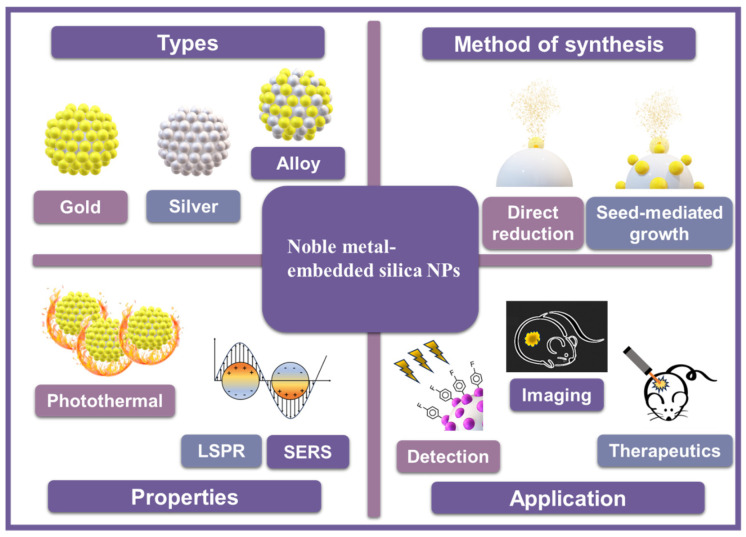
Summary of topics related to nanoparticle synthesis methods, types, characteristics, and application areas such as detection, therapeutics, and imaging.

**Figure 2 nanomaterials-14-00268-f002:**
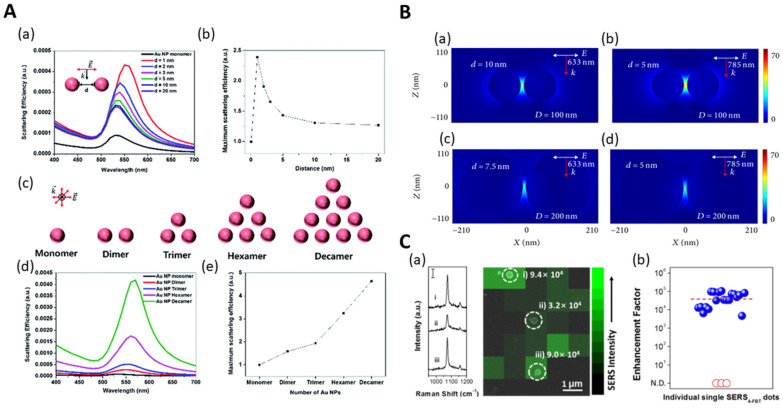
(**A**) Simulating scattering efficiency of various Au NPs using DDA calculation. (**a**) Theoretical limit efficiency spectrum depending on distance between two mono AuNPs. (**b**) Maximum value of theoretical scattering efficiency spectrum. (**c**) Schematic diagram of theoretical scattering efficiency of assembled Au NPs. (**d**) Theoretical scattering efficiency spectrum of assembled Au NPs. (**e**) Scattering efficiency divided by number of AuNPs in I_max_ of assembled Au NP structure, adapted from Lee et al., 2015 [31]. (**B**) FDTD simulation images of Ag nanoshells with different diameters and shell thicknesses at 633 nm (**a**,**c**) and 785 nm (**b**,**d**) light sources, adapted from Zhao et al. [41]. (**C**) (**a**) SERS intensity map of single SERS dot processed via 4-FBT in SEM image. (**b**) Single SERS dot 23 SERS enhancement factors. The red line represents the average SERS enhancement factor. adapted from Kim et al., 2016 [42].

**Figure 3 nanomaterials-14-00268-f003:**
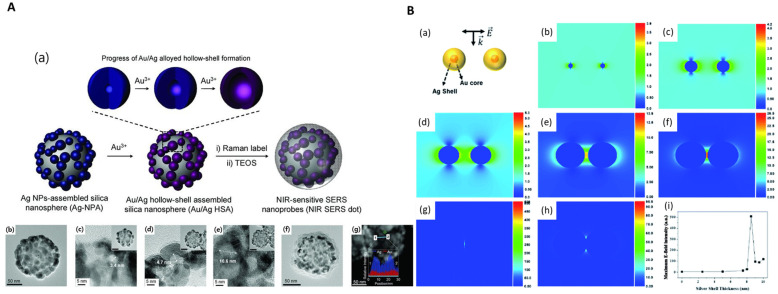
(**A**) (**a**) Synthesis scheme of Au/Ag alloyed hollow-shell. TEM images of (**b**) AuNHs, (**c**–**e**) hollow structure of AuNHS, and (**f**) AuNHS@SiO_2_. (**g**) Atomic profiling of AuHNS, adapted from Kang H et al., 2013 [56]. (**B**) (**a**) Simulation model of interparticle distance between two Au@Ag NPs, (**b**) 3 nm Au NP core dimer, (**c**–**h**) Ag shell thickness of Au@Ag NPs (**c**) 2.5 nm, (**d**) 5 nm, (**e**) 7.5 nm, (**f**) 8 nm, (**g**) 8.5 nm, and (**h**) 9 nm. (**i**) Maximum E-field intensity between Au NPs and Au@Ag NPs with Ag shell thickness from 2.5 nm to 10 nm, adapted from Pham et al., 2017 [52].

**Figure 4 nanomaterials-14-00268-f004:**
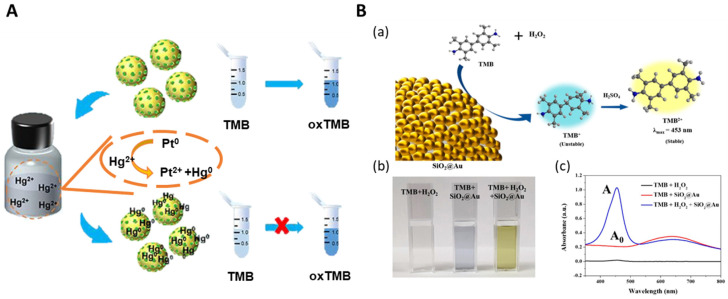
(**A**) Catalytic activity scheme of AuPt@DSN using dendritic silica nanosphere (DSN) as the core, adapted from Zhou J et al., 2022 [64]. (**B**) (**a**) Scheme of peroxidase-like activity of SiO_2_@Au NPs. (**b**) Optical images. (**c**) UV-Vis absorbance spectroscopy with TMB, with H_2_O_2_ (A), and without H_2_O_2_ (A_0_), adapted from Seong B et al., 2021 [34].

**Figure 5 nanomaterials-14-00268-f005:**
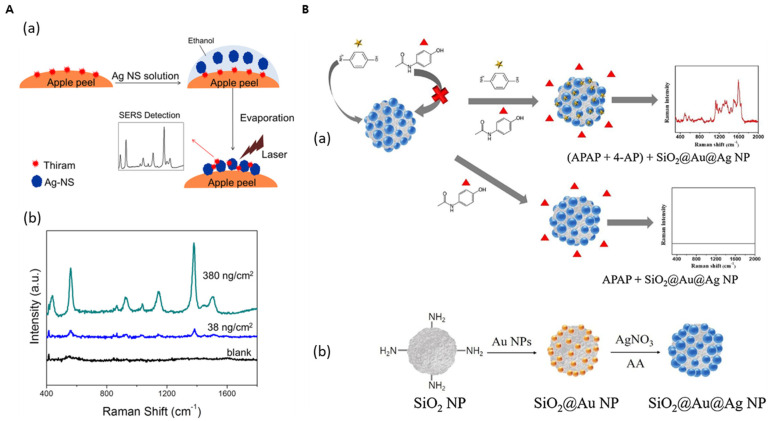
(**A**) (**a**) Schematic illustration of thiram detection in apple peel utilizing AgNS. (**b**) Raman intensity by concentration of thiram. Adapted from Yang J et al., 2014 [68]. (**B**) (**a**) Schematic of detection of 4-aminophenol (4-AP, ☆) as impurity in acetaminophen (APAP; ▲) via surface-enhanced Raman scattering on Au-Ag alloy embedded silica nanoparticles (SiO_2_@Au@Ag NPs). (**b**) Synthesis of SiO_2_@Au@Ag NPs. Adapted from Pham et al., 2020 [67].

**Figure 7 nanomaterials-14-00268-f007:**
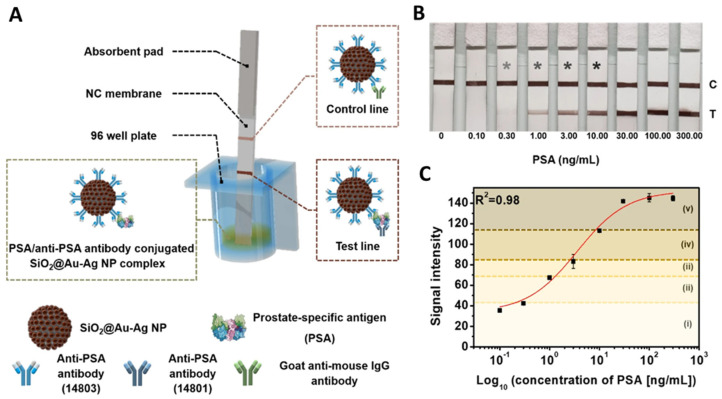
(**A**) Schematic illustration of PSA detection using LFIA platform with SiO_2_@Au-Ag NPs as a signal reporter. (**B**) Color change depending on PSA concentration (*: Can be checked with the naked eye). (**C**) Schematic of early diagnosis and prognosis detection for prostate cancer by measuring signal intensity (i: no recurrence, ii: recurrence, iii: no cancer, iv: early-stage disease, v: late-stage disease), adapted from Kim H-M et al., 2021 [89].

**Figure 8 nanomaterials-14-00268-f008:**
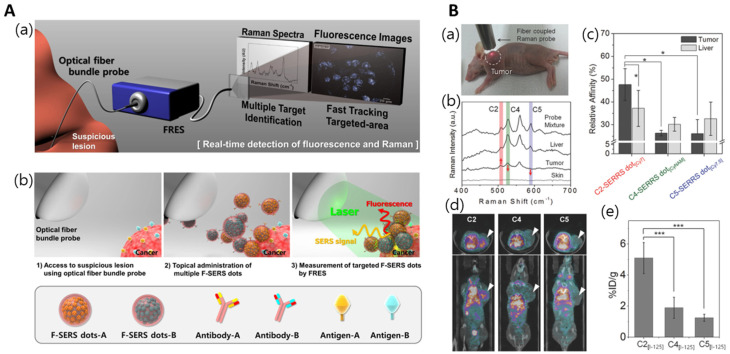
(**A**) (**a**) Dual-modal detection with fluorescence and Raman scattering. (**b**) Illustration of in vivo multiplexed molecular imaging procedure, adapted from Jeong S et al., 2015 [94]. (**B**) (**a**) Photograph of SERS measurement with fiber-coupled portable Raman system at 1 h post-intravenous injection. (**b**) SERS spectra of the skin, liver, and tumor sites of the mouse. (**c**) Ratiometric quantitative analysis of in vivo relative affinity screening of C2-SERRS dot[Cy7], C4-SERRS dot[CyNAM], and C5-SERRS dot[Cy7.5] (* *p* < 0.05). (**d**) SPECT/CT images of human colon cancer xenograft mice at 3 h post-injection of I-125-labeled antibody. (**e**) Quantitative analysis of in vivo affinity screening for three I-125-labeled antibodies by SPECT/CT images at tumor sites of each mouse (*** *p* < 0.001), adapted from Kang H et al., 2018 [95].

**Figure 9 nanomaterials-14-00268-f009:**
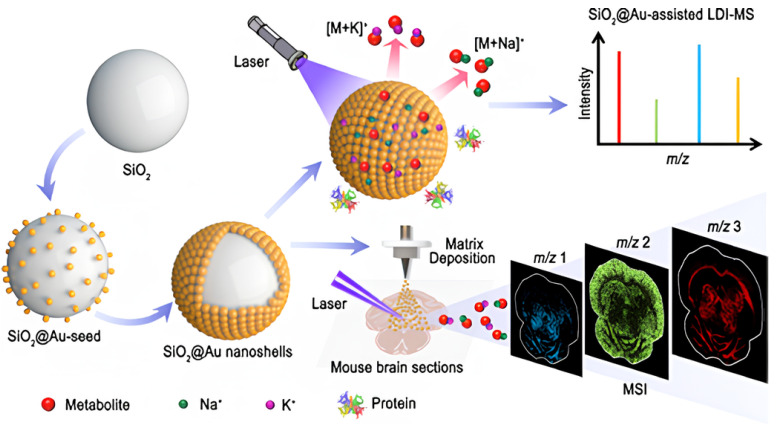
Schematic illustration of LDI-MS and MSI utilizing SiO_2_@Au nanoshells as an effective matrix for the analysis of small molecules. Small molecules trapped on the SiO_2_@Au nanoshell surface easily form sodium/potassium adduct ions during laser irradiation, adapted from Du M et al., 2022 [96].

**Figure 10 nanomaterials-14-00268-f010:**
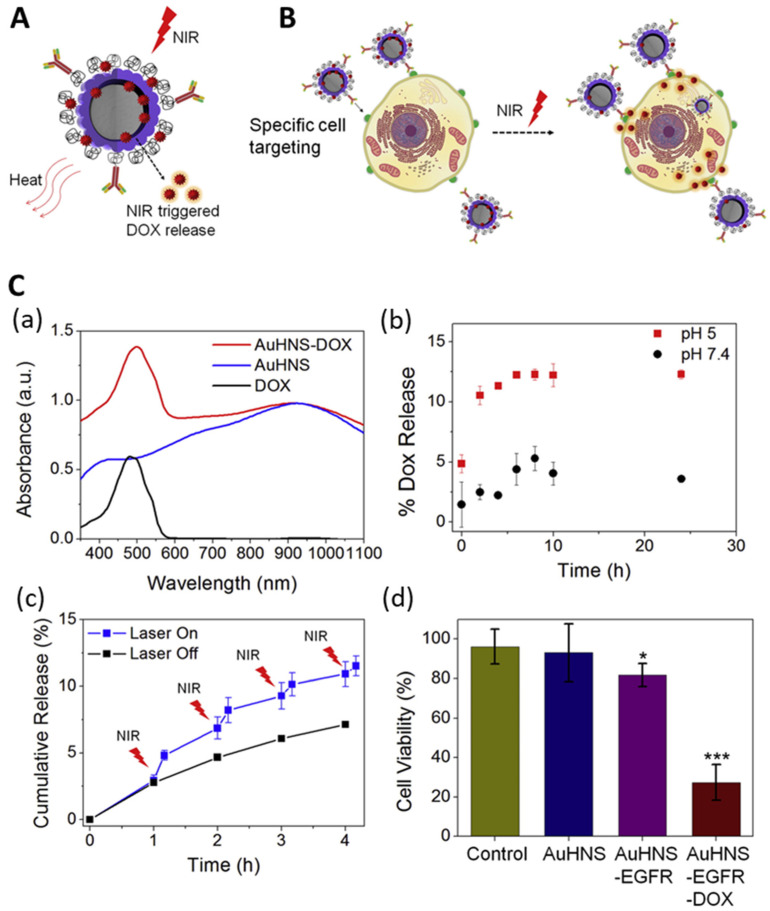
(**A**,**B**) Illustration to explain functions of DOX-loaded AuHNS complexes. (**A**) NIR light-triggered doxorubicin release and heat generation and (**B**) specific cancer cell targeting. (**C**) (**a**) UV-Vis spectra of AuHNS-EGFR-DOX, AuHNS, and free DOX indicating loading of DOX to AuHNS. (**b**) Doxorubicin release profile of AuHNS-DOX at different pH. (**c**) Induced release of doxorubicin from AuHNS-DOX by NIR irradiation (800 nm) at pH 7.4. (**d**) Quantitative cell viability results of A549 cells treated with different AuHNS complexes for 48 h (*: *p* < 0.05, ***: *p* < 0.001), adapted from Noh M.S. et al., 2015 [57].

**Figure 11 nanomaterials-14-00268-f011:**
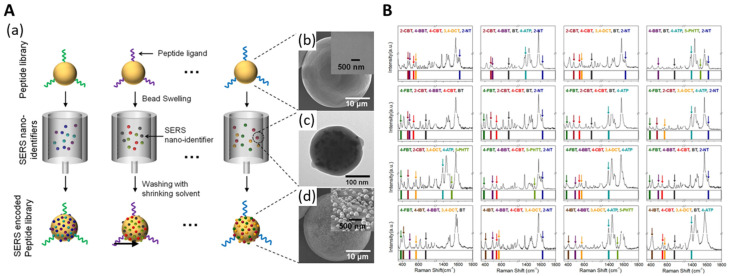
(**A**) Schematic of peptide-encoding process with SERS IDs and electron microscopic images at each step. (**a**) Peptide-encoding process by attaching SERS IDs. (**b**) Field emission scanning electron microscope (FE-SEM) images of microbeads without encoding. (**c**) TEM image of SERS ID comprised Ag NPs embedded in silica nanosphere. (**d**) FE-SEM images of microbeads with SERS encoding. (**B**) Sixteen representative SERS spectra and their corresponding barcode presentations for microbeads encoded with five SERS ID combinations, adapted from Kang H et al., 2015 [103].

**Table 2 nanomaterials-14-00268-t002:** Summary of nanozyme.

Metal-Embedded Silica NP	Catalyst	Optimization of Catalytic Performance	Results	Application	Reference
SiO_2_@Au@Au	Peroxidase	1. TMB conc. 2. H_2_O_2_ conc. 3. pH solution 4. NP amount 5. Reaction time 6. Termination time	1. 0.8 mM 2. 200 mM 3. pH 4 4. 20, 25 mg 5. 25 min 6. 5 min	-	[34]
SiO_2_@Au@Pt	Peroxidase	1. NP amount 2. TMB conc. 3. Incubation time 4. pH solution	1. 5 μg 2. 0.5 mM 3. 15 min 4. pH 4	-	[53]
AuPt@DSN	Peroxidase	1. TMB conc. 2. H_2_O_2_ conc.	1. 0.4 mM 2. 4 mM	Hg^+^ detection	[64]
SiO_2_@Au@Ag	Peroxidase	1. TMB conc. 2. Incubation time 3. NP amount 4. pH solution	1. 0.8 mM 2. 15 min 3. 20 μg 4. pH 6	H_2_O_2_ detection	[65]

Abbreviations: Nanoparticle (NP); 3,3′,5,5′-Tetramethylbenzidine (TMB) dendritic silica nanoparticle (DSN)

**Table 3 nanomaterials-14-00268-t003:** Summary of sensing and detection.

Metal-Embedded Silica NP	Modification of NPS	Detection Method	Target Material	LOD	Reference
SiO_2_@Ag	Anti-PSA antibody	SIA	PSA	2.0 pg/mL	[44]
SiO_2_@Ag	Molecular imprinted polymers	SERS	ofloxacin	2.7 × 10^−9^ M	[49]
SiO_2_@Au@Ag	-	SERS	Histamine	3.698 ppm	[66]
SiO_2_@Au@Ag	-	SERS	4-AP	3.5 ppm	[67]
Ag NS	-	SERS	Thiram	38 ng/cm^2^	[68]
SiO_2_@Ag_4-FBT_	-	SERS	Hg^+^	0.819 μM	[69]
SiO_2_@Au@Ag	4-MPBA	SERS	Glucose	1. 0.15 mM	[71]
SiO_2_@Au CSNP	SARS-CoV-2 nucleocapsid protein antibody	LFIA	SARS-CoV-2 nucleocapsid protein	0.24 pg/mL	[74]
SiO_2_@AgSiO_2_	PSA capture antibody	SIA	PSA	0.11 pg/mL	[79]
SiO_2_@Au@Ag	-	Liposome decomposition SIA	4-ATP	1.3 × 10^−17^ mol	[82]
SiO_2_@Au seed	-	Colorimetric immunoassay	IgG	0.021 ng/mL	[85]
SiO_2_@Au@Ag	Anti-PSA antibody	LFIA	PSA	0.2 ng/mL	[89]

Abbreviations: 4-mercaptophenyl boronic acid (4-MPBA), surface-enhanced Raman scattering (SERS), SERS-based immunoassay (SIA), and lateral flow immunoassay (LFIA).

**Table 4 nanomaterials-14-00268-t004:** Summary of bioimaging.

Metal-Embedded Silica NP	a. RLC b. Fluorescence Dye	Ligand	Specific Target	Imaging Method	Reference
SiO_2_@Ag	a. 4-ATP, 4-MT b. FITC, AF647	Annexin V	phosphatidylserine	Fluorescence SERS	[91]
SiO_2_@Ag	a. RITC, FITC b. AF610	Anti-HER2 Anti-EGFR	MDA-MB-231/HER2 breast cancer cell	Fluorescence SERS	[94]
Au-Ag hollow shell	a. Cy7LA a. CyNAMLA a. Cy7.5LA	C2 antibody	TSPAN8	NIR-SERRS	[95]
SiO_2_@Au	-	-	Strawberry zebrafish, honeybee, mouse brain tissues	LDI-MS	[96]

Abbreviations: 4-aminothiolphenol (4-ATP), 4-mercaptotoluene (4-MT), fluorescein isothiocyanate (FITC), Alexafluoro 647 (AF647), rhodamine B isothiocyanate (RITC), Alexa Fluor 610-X (AF610), Human epidermal growth factor receptor 2 (HER2), epidermal growth factor receptor (EGFR), tetraspanin-8 (TSPAN8), near infrared-active surface-enhanced resonance Raman scattering (NIR-SERRS), and laser desorption/ionization mass spectrometry (LDI-MS).

**Table 5 nanomaterials-14-00268-t005:** Summary of drug carriers and photothermal therapy.

Metal-Embedded Silica NP	Ligand	Cancer Therapy Method	Specific Target	Cell Viability	Reference
SiO_2_@Au@GO	-	Photothermal effect Docetaxel	DU145 cells	37%	[38]
SGS	-	Photothermal effect	hMSC	-	[40]
AuHNs	Anti-EGFR	Photothermal effect Doxorubicin	A549 cells	35%	[57]
SiO_2_@Ag	Cysteinyl-β-CD EDA-β-CD	Doxorubicin	MCF-7 cells	60%	[102]

Abbreviations: human mesenchymal stem cells (hMSC), adenocarcinomic human alveolar basal epithelial cells (A549) and ethylenediamine-β-CD (EDA-β-CD).

## Data Availability

The data presented in this study are available on request from the corresponding author. Graphical content was partially provided by Freepik.com (https://www.freepik.com).

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
