# Peer review of "Recent Studies on Metal-Embedded Silica Nanoparticles for Biological Applications"

_nanomaterials, 2024, doi:10.3390/nano14030268_

Round 1
Reviewer 1 Report
Comments and Suggestions for Authors
This review provides a survey on several unique designs and applications of metal-embedded silica NPs in recent years. I think this article is appropriate for nanomaterials. I recommend posting it with some minor modifications.
1. This article gives some pictures and tables when introducing the spectroscopic applications of metal nanoparticles embedded in silica nanoparticles. However, the contents in the table lack detailed introduction in the text, which is not conducive to readers' understanding of the research work listed in the table. It is recommended that the author briefly introduce the research work shown in the table below in the article.
2. In section 3.1, the advantages of nanoparticles as nanozymes are introduced. It is also necessary to further introduce the specific applications and achievements of these advantages in fuel cells, hydrogenation, air purification, anti-aging treatment, and cancer treatment.
3. Section 3.4 introduces photothermal therapy. It is also necessary to introduce how metal nanoparticles are embedded in silica nanoparticles to achieve the photothermal effect and what role it plays.
4. The authors also need to add in the concluding remarks the future prospects of metal nanoparticles embedded in silica nanoparticles in the biological and medical fields.
Comments on the Quality of English LanguageNo comments.
Reviewer 2 Report
Comments and Suggestions for Authors
Dear authors, I read with interest your manuscript entitled "Recent Studies on Metal Nanoparticles Embedded Silica Nanoparticles for Spectroscopic Applications". I recommend the manuscript to be published after major revision. I have some suggestions/comments/questions that may contribute to the work:
- I recommend rewriting the article title, "Recent Studies on Noble Metal-Embedded Silica Nanoparticles for Biological Applications"
-L67-L68 "2. Optical Properties by Structure and Composite of Noble Metal-Embedded Silica 67 NPs"/ L161 "Spectroscopic Applications of Metal-Embedded Silica NPs" - the section title not provide a clear and concise indication of the content that follows.
- -Authors asked for the permission to (re)use the images?
- Table 1.A new column with references should be added, Silica Core (nm): only values, Metal NP (nm) only values in the table
The approach/mechanism involved in the prezented applications should be detailed.
I suggest the authors to add one table after each subsection: Nanozyme, Sensing and detection, Bioimaging, Drug Carriers and Photothermal Therapy in order to highlight the main information, as well to perform a comparative study between different NPs, discussing advantages and disadvantages.
Reviewer 3 Report
Comments and Suggestions for Authors
Authors based on literature discussed application of metal-embedded silica NPs in various spectroscopic serving. The metal-embedded silica NPs as nanozymes, detection and imaging probes, drug carriers, photothermal inducers, and bioactivation molecule screening identifiers are presented. The literature review is prepared logically, taking into account relevant original works from the recent period. The authors presented the use of modified silica in various aspects without assessing their strengths and weaknesses. Review papers should also present a critical or enthusiastic assessment of the problem. Such a discussion should be added in the publication. The use of modified silicas is associated with their strong impact on the environment. Sustainable development requires also presenting in review works the toxicological effects of the use of new materials on the environment and people. This discussion should also be added to the manuscript.
Publications are recommended for publication in the journal after introducing a discussion about the limitations in the use of new materials into the text.
Round 2
Reviewer 2 Report
Comments and Suggestions for Authors
The authors have addressed all the comments by reviewers, and thus I recommend its publication in this journal without further revision.